# An observational treatment study of metacognition in anxious-depression

Celine Ann Fox[1,2]*, Chi Tak Lee[1,2], Anna Kathleen Hanlon[1,2], Tricia XF Seow[3], Kevin Lynch[1], Siobhán Harty[4], Derek Richards[1,4], Jorge Palacios[1,4], Veronica O'Keane[5,6], Klaas Enno Stephan[7,8], Claire M Gillan[1,2,9]

[1]School of Psychology, Trinity College Dublin, Dublin, Ireland; [2]Trinity College Institute of Neuroscience, Trinity College Dublin, Dublin, Ireland; [3]Wellcome Centre for Human Neuroimaging, University College London, London, United Kingdom; [4]SilverCloud Science, SilverCloud Health Ltd, Dublin, Ireland; [5]Department of Psychiatry, Trinity College Dublin, Dublin, Ireland; [6]Tallaght Hospital, Trinity Centre for Health Sciences, Tallaght University Hospital, Dublin, Ireland; [7]Translational Neuroimaging Unit (TNU), Institute for Biomedical Engineering, University of Zurich, Zurich, Switzerland; [8]Max Planck Institute for Metabolism Research, Cologne, Germany; [9]Global Brain Health Institute, Trinity College Dublin, Dublin, Ireland

*For correspondence:
foxce@tcd.ie

**Abstract** Prior studies have found metacognitive biases are linked to a transdiagnostic dimension of anxious-depression, manifesting as reduced confidence in performance. However, previous work has been cross-sectional and so it is unclear if under-confidence is a trait-like marker of anxious-depression vulnerability, or if it resolves when anxious-depression improves. Data were collected as part of a large-scale transdiagnostic, four-week observational study of individuals initiating internet-based cognitive behavioural therapy (iCBT) or antidepressant medication. Self-reported clinical questionnaires and perceptual task performance were gathered to assess anxious-depression and metacognitive bias at baseline and 4-week follow-up. Primary analyses were conducted for individuals who received iCBT (n=649), with comparisons between smaller samples that received antidepressant medication (n=82) and a control group receiving no intervention (n=88). Prior to receiving treatment, anxious-depression severity was associated with under-confidence in performance in the iCBT arm, replicating previous work. From baseline to follow-up, levels of anxious-depression were significantly reduced, and this was accompanied by a significant increase in metacognitive confidence in the iCBT arm ($\beta$=0.17, SE=0.02, p<0.001). These changes were correlated (r(647)=-0.12, p=0.002); those with the greatest reductions in anxious-depression levels had the largest increase in confidence. While the three-way interaction effect of group and time on confidence was not significant (F(2, 1632)=0.60, p=0.550), confidence increased in the antidepressant group ($\beta$=0.31, SE = 0.08, p<0.001), but not among controls ($\beta$=0.11, SE = 0.07, p=0.103). Metacognitive biases in anxious-depression are state-dependent; when symptoms improve with treatment, so does confidence in performance. Our results suggest this is not specific to the type of intervention.

## eLife assessment

This **valuable** study advances our knowledge of the effects of anxiety/depression treatment on metacognition, demonstrating that treatment increases metacognitive confidence alongside improving symptoms. The authors provide **convincing** evidence for the state-dependency of metacognitive confidence, based on a large longitudinal treatment dataset. However, it is unclear to what extent this effect is truly specific to treatment, as changes in metacognitive confidence in the group receiving online therapy were not statistically different from those in the control group.

## Introduction

Metacognition refers to the ability to accurately monitor and appraise one's own cognitive experience (*Fleming and Lau, 2014*). Metacognition is crucial for adaptive behaviour: it allows for flexible adjustment of behavioural strategies in order to improve performance, signals when to engage or withdraw from an activity, and guides the engagement in social interactions (*Fleming and Daw, 2017*; *Fleming et al., 2012*). Metacognitive abilities vary across individuals; one can be under- or over-confident, and these biases can be associated with maladaptive thoughts, feelings and behaviours (*Philipp et al., 2020*). There is a growing interest in these confidence abnormalities in psychiatry, with studies implicating alterations of metacognition in depression, obsessive-compulsive disorder, and psychosis (*Hoven et al., 2019*). While case-control studies have mainly found patterns of reduced confidence across several disorders, newer methods that can separate transdiagnostic dimensions of mental health using large online samples have revealed specific and bi-directional effects of confidence (*Wise et al., 2023*). Using these methods, studies have shown that the transdiagnostic dimension 'anxious-depression' is linked to under-confidence in one's own performance, while a separate dimension 'compulsivity and intrusive thought' is related to elevated confidence (*Rouault et al., 2018*; *Seow and Gillan, 2020*; *Hoven et al., 2022*; *Seow et al., 2021*; *Benwell et al., 2022*). A major gap in this area is that studies to-date only measure metacognition and transdiagnostic psychopathology at a single time point. Therefore, it is unclear if metacognitive biases are stable, fixed traits, or if they might change with treatment response. Preliminary evidence suggests metacognition may indeed be malleable; metacognitive abilities can be improved with metacognitive interventions, such as training, in unselected online samples (*Carpenter et al., 2019*; *Engeler and Gilbert, 2020*) and in clinical populations (*Jelinek et al., 2017*; *Lysaker et al., 2018*). However, it remains unknown if metacognitive changes generalise beyond its specific training context and are associated with any real-world improvement in psychiatric symptoms. In clinical studies, research has identified confidence abnormalities in at-risk populations (*Eisenacher et al., 2015*; *Gawęda et al., 2018*), suggestive of a trait-dependence. In contrast, stimulant use disorders remitters have better metacognition than active users, suggesting state-dependence (*Moeller et al., 2016*). Within-subject designs are needed to extend this work and understand if metacognition can improve in parallel to symptom alleviation, or if those with greater metacognitive deficits are simply the most vulnerable to illness onset and persistence. The present study aimed to address this by examining metacognition in a large cohort of individuals before and after internet-based cognitive behavioural therapy (iCBT). iCBT has emerged as an important intervention for reducing the treatment-gap in mental healthcare provision globally; it is low-cost, scalable, geographically unconstrained and flexible (*Mogoaşe et al., 2017*; *Webb et al., 2017*). iCBT offers patients standardised content and records objective metrics of treatment engagement, making it particularly well-suited to treatment-oriented research in psychiatry (*Lee et al., 2023*). Additionally, iCBT has demonstrated clinical effectiveness in terms of symptom improvement (*Andersson et al., 2019*; *Eilert et al., 2021*; *Karyotaki et al., 2021*). While one study found that iCBT modified self-reported metacognitive beliefs (*Newby et al., 2014*), it remains unknown if metacognitive confidence in decision-making improves following successful iCBT. In the current study, we used an objective task measure of metacognition (*Fleming et al., 2014*), which allowed us to test if successful treatment is linked to within-person improvements in metacognition. We also tested if any changes in metacognition were iCBT-specific, by comparing data gathered from smaller samples of individuals receiving antidepressant medication and a control group receiving no intervention. Similar to iCBT, antidepressants have established transdiagnostic efficacy (*Cipriani et al., 2018*; *Gøtzsche and Dinnage, 2020*; *Skapinakis et al., 2016*). However, studies examining the impact of antidepressants on cognition have typically focused on cognitive capacities other than metacognition (*Prado et al., 2018*; *Rosenblat et al., 2015*; *Salagre et al., 2017*; *Vernon et al., 2014*). Accordingly, a secondary aim of this study was to compare metacognitive changes across the different intervention arms, which may shed light on differential therapeutic mechanisms and potentially augment therapeutic decision-making in the future.

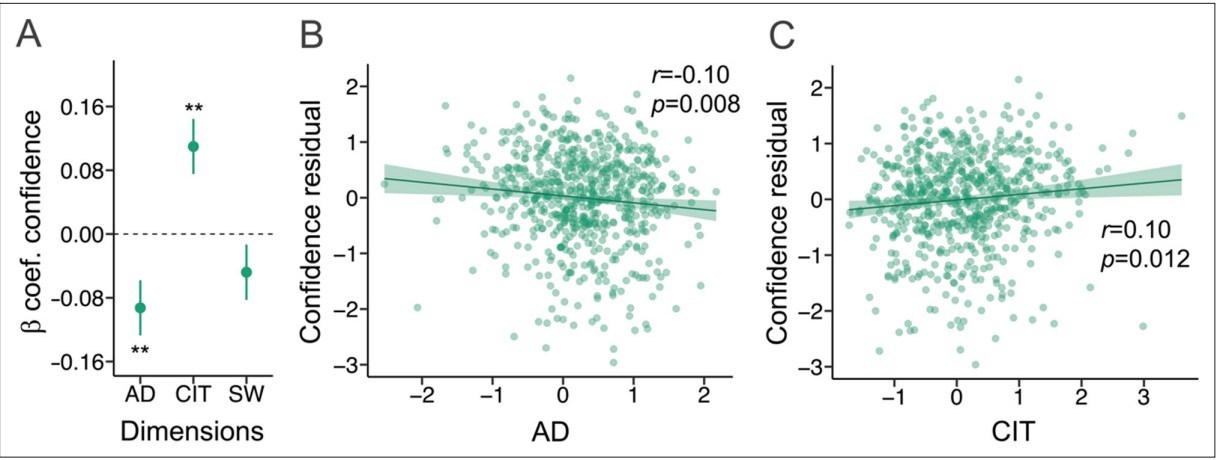

**Figure 1.** Cross-sectional findings at baseline in the iCBT arm. *β*=standardised beta coefficient, *r*=correlation coefficient, *p*=p-value, AD = Anxious-Depression, CIT = Compulsivity and Intrusive Thought, SW = Social Withdrawal. The error bars represent the standard error around the standardised beta coefficient. N=649. (**A**) AD and CIT were associated with metacognitive bias, while SW was not, using linear regression analysis. (**B**) The residual values for confidence (controlling for age, gender and education) were negatively correlated with AD. (**C**) The residual values for confidence (controlling for age, gender and education) were positively correlated with CIT.

## Results

### Cross-sectional findings at baseline: iCBT

At baseline, age was not significantly associated with mean confidence ($\beta$=0.003, SE = 0.003, p=0.401). Males had significantly higher confidence (M=3.91, SD = 0.81), than females (M=3.75, SD = 0.86) ($\beta$=0.08, SE = 0.04, p=0.044). Mean confidence was significantly lower among those with educational attainment above undergraduate level (M=3.73, SD = 0.89), when compared to participants with attainment below undergraduate (M=3.94, SD = 0.87) ($\beta$=0.21, SE = 0.10, p=0.036). Mean confidence did not significantly differ when comparing those who had educational attainment above undergraduate (M=3.73, SD = 0.89) to those who had some or had completed an undergraduate degree (M=3.74, SD = 0.83) ($\beta$=0.01, SE = 0.08, p=0.921). When comparing task measures, mean confidence was not significantly associated with mean accuracy ($\beta$=−1.29, SE = 1.42, p=0.366), mean dot difference (task difficulty) ($\beta$=0.002, SE = 0.002, p=0.406), or mean response time ($\beta$=6.47, SE = 15.13, p=0.669).

Participants with higher levels of anxious-depression had lower levels of mean confidence ($\beta$=−0.09, SE = 0.03, p=0.008; *Figure 1A and B*), while those with higher levels of compulsivity and intrusive thought had elevated mean confidence ($\beta$=0.11, SE = 0.03, p=0.002; *Figure 1A and C*), controlling for age, gender, and education. Levels of social withdrawal were not associated with mean confidence ($\beta$=−0.05, SE = 0.03, p=0.168; *Figure 1A*).

### Treatment findings: iCBT

The transdiagnostic dimensions and psychiatric scale scores all significantly improved from baseline to four-week follow-up, except for impulsivity (*Figure 2A* and *Figure 2—figure supplement 1*). In tandem with these clinical changes, there was a small but significant increase in mean confidence from baseline (M=3.78, SD=0.85) to follow-up (M=3.95, SD=0.89), ($\beta$=0.17, SE = 0.02, p<0.001, $r^2$=0.01) (*Figure 2B*). Although mean accuracy did not change from baseline (M=0.71, SD=0.02) to follow-up (M=0.71, SD = 0.02) ($\beta$=0.001, SE = 0.001, p=0.299), due to the staircasing procedure, participants' ability to detect differences between the visual stimuli improved. This was reflected as the overall increase in task difficulty to maintain the accuracy rates from baseline (dot difference: M=41.82, SD=11.61) to follow-up (dot difference: M=39.80, SD=12.62), ($\beta$=−2.02, SE = 0.44, p<0.001, $r^2$=0.01) (*Figure 2C*). Additionally, the effect of time on confidence was not dependent on how much participants engaged in iCBT, as indexed by time spent in the program ($\beta$<0.01, SE <0.01, p=0.756) and percentage of the iCBT program viewed ($\beta$=0.09, SE = 0.21, p=0.650). Confidence changes did not differ across the different types of iCBT programs, when the Space from Depression and Anxiety program was compared pairwise to Space from Depression ($\beta$=−0.09, SE = 0.17, p=0.594), Space

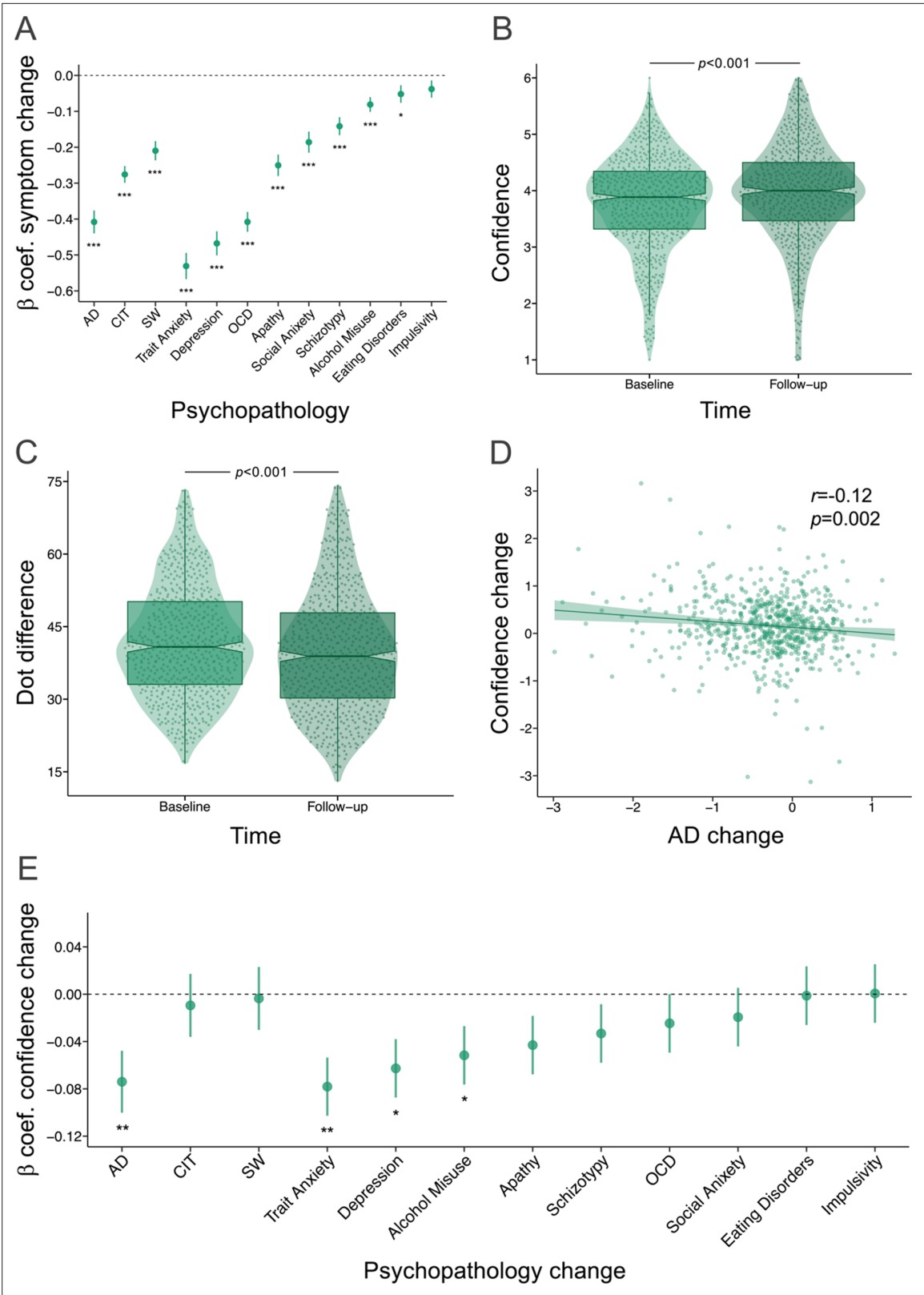

**Figure 2.** Treatment findings in the iCBT arm. *β*=standardised beta coefficient, AD = Anxious-Depression, CIT = Compulsivity and Intrusive Thought, SW = Social Withdrawal, OCD = Obsessive compulsive disorder, r=correlation coefficient, p=p-value (unadjusted), ***=$p < 0.001$, **=$p < 0.01$, *=$p < 0.05$. The error bars represent the standard error around the standardised beta coefficient. Regression analyses were used for all tests. N=649. (**A**) Psychopathology symptoms improved with four weeks of iCBT. (**B**) Confidence was significantly higher and, (**C**) the task was more difficult at 4-

*Figure 2 continued on next page*

*Figure 2 continued*

week follow-up. (**D**) Those with the largest improvements in AD had the greater increases in confidence. (**E**) Change in confidence also scaled with improvements in trait anxiety, depression and alcohol misuse.

The online version of this article includes the following figure supplement(s) for figure 2:

**Figure supplement 1.** Changes in psychiatric dimensions and scale scores from baseline to follow-up in the iCBT arm (N=649) using regression analyses.

**Figure supplement 2.** The interaction effect of time and psychiatric dimension/scale change on mean confidence in the iCBT arm (N=649) using regression analyses.

from Anxiety ($\beta$=−0.15, SE = 0.17, p=0.376), Life Skills ($\beta$=−0.15, SE = 0.19, p=0.435) or the 'other' category comprising miscellaneous iCBT programs ($\beta$=−0.03, SE = 0.18, p=0.881). Change in confidence was not different among those receiving concurrent treatment versus not ($\beta$=−0.03, SE = 0.06, p=0.566). To test if changes in confidence from baseline to follow-up scaled with changes in anxious-depression, we ran a repeated measure regression analyses with per-person changes in anxious-depression as an additional independent variable. We found this was the case, evidenced by a significant interaction effect of time and change in anxious-depression on confidence ($\beta$=−0.12, SE = 0.04, p=0.002). Those with the largest decrease in anxious-depression had the greatest increase in confidence. This was similarly evident in a simple correlation between change in confidence and change in anxious-depression (r(647)=-0.12, p=0.002) (*Figure 2D*). This effect was specific to anxious-depression; the interaction effect of time and change in compulsivity and intrusive thought on mean confidence was not significant ($\beta$=−0.06, SE = 0.05, p=0.221). Similarly, the significant interaction effect of time and anxious-depression on mean confidence held when including change in the other transdiagnostic dimensions (compulsivity and intrusive thought and change in social withdrawal) as covariates in the model ($\beta$=−0.07, SE = 0.03, p=0.005). The interaction effect of time and change in anxious-depression on task difficulty was not significant ($\beta$=0.14, SE = 0.69, p=0.835). To test the extent to which baseline differences in mean confidence or anxious depression might drive the results, we re-ran these regression analyses with baseline measures instead of change indices. There was no significant interaction effect of time and baseline confidence ($\beta$=0.06, SE = 0.04, p=0.144) or an interaction effect of time and baseline anxious-depression ($\beta$=−0.03, SE = 0.05, p=0.611) on mean confidence. Exploratory analyses further tested the specificity of these effects to anxious-depression by examining the interaction effect of time and change in each psychiatric score on mean confidence. Changes in trait anxiety ($\beta$=−0.08, SE = 0.02, p=0.002), depression ($\beta$=−0.06, SE = 0.02 p=0.011) and alcohol misuse ($\beta$=−0.05, SE = 0.02, p=0.037) also showed an association with changes in confidence (*Figure 2E* and *Figure 2—figure supplement 2*).

## Comparing iCBT, antidepressant and control groups

When comparing the three groups directly, ANOVA analysis predicting anxious-depression scores with group and time as independent variables revealed a main effect of time (F(1, 1632)=62.99, p<0.001), a main effect of group (F(2, 1632)=249.74, p<0.001), and an interaction effect of group and time (F(2, 1632)=9.23, p<0.001). Examining simple effects in the antidepressant arm, there was a significant reduction in anxious-depression from baseline to follow-up ($\beta$=−0.61, SE = 0.09, p<0.001). Among controls, levels of anxious-depression did not significantly change ($\beta$=0.10, SE = 0.06, p=0.096). Further details of transdiagnostic clinical changes for the antidepressant and controls groups are presented in *Figure 3A* and *Figure 3—figure supplement 1*. Predicting confidence scores using ANOVA analysis with group and time as independent variables revealed a main effect of time (F(1, 1632)=16.26, p<0.001), and no significant main effect of group (F(2, 1632)=2.35, p=0.096). The interaction effect of group and time on mean confidence was not significant (F(2, 1632)=0.60, p=0.550), suggesting that change in confidence did not differ across the three groups. Tests of simple effects revealed that mean confidence significantly increased from baseline (M=3.77, SD=0.88) to follow-up (M=4.07, SD=0.79) in the antidepressant arm ($\beta$=0.31, SE = 0.08, p<0.001) (*Figure 3B*). Among controls, there was no significant change in confidence from baseline (M=3.68, SD=0.86) to follow-up (M=3.79, SD=0.92) ($\beta$=0.11, SE = 0.07, p=0.103) (*Figure 3B*). With respect to task performance, there was a significant main effect of time (F(1, 1632)=15.17, p=0.001) and group (F(2, 1632)=4.56, p=0.011) on mean dot difference when the three groups were included in the model. The interaction effect of time and group on mean

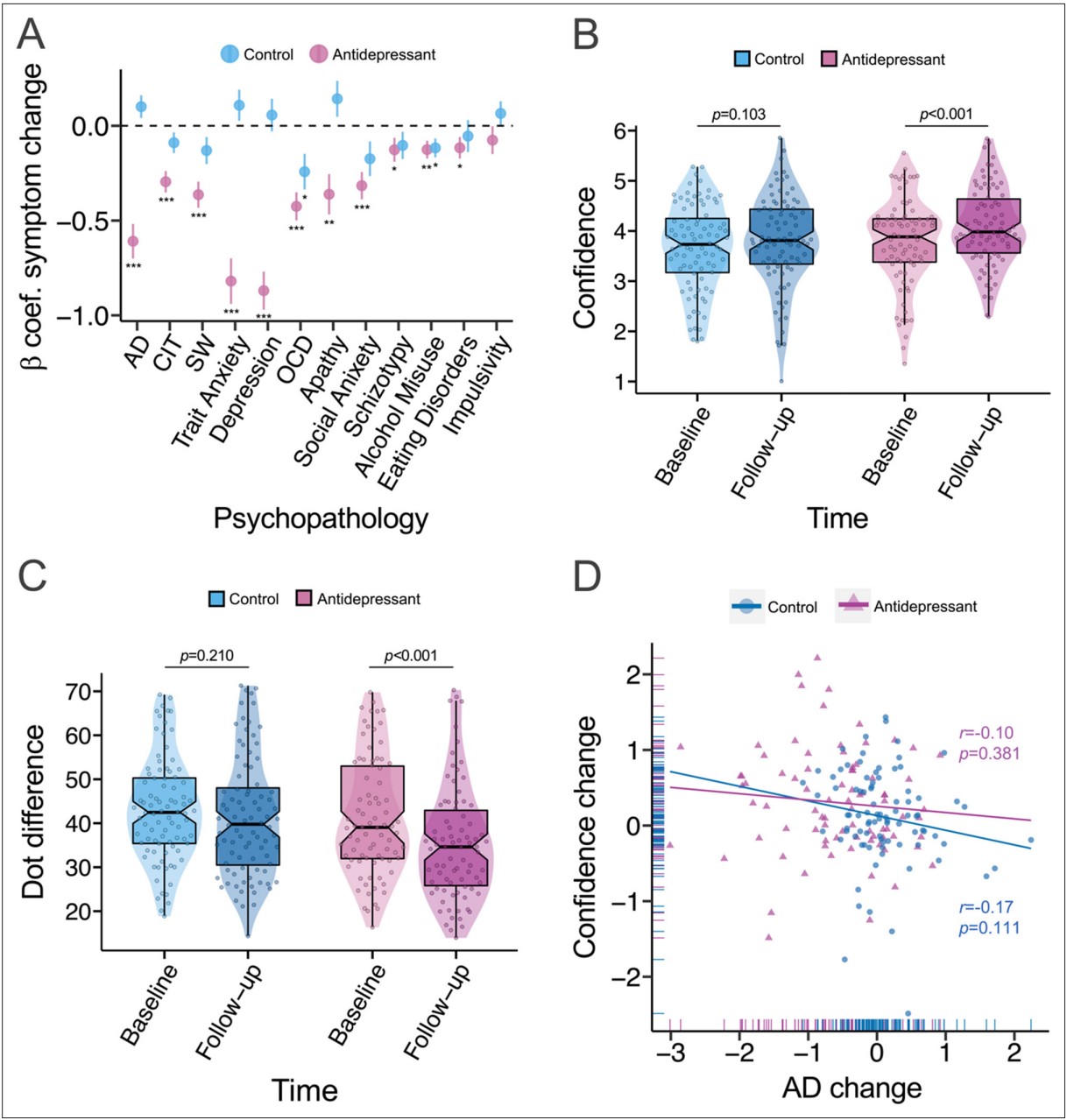

**Figure 3.** Comparing iCBT, antidepressant and control groups. *β*=standardised beta coefficient, AD = Anxious-Depression, CIT = Compulsivity and Intrusive Thought, SW = Social Withdrawal, OCD = Obsessive compulsive disorder, *r*=correlation coefficient, *p*=p-value, ***=p < 0.001, **=p < 0.01, *=p < 0.05. The error bars represent the standard error around the standardised beta coefficient. Regression analyses were used for tests. (**A**) The majority of psychiatric scales improved in the antidepressant arm (N=82) after 4 weeks of treatment, while the controls (N=88) only had significant reductions in OCD symptoms and alcohol misuse at follow-up. (**B**) While confidence increased in the antidepressant arm, there was no significant change in confidence among controls. The larger increase in confidence in the antidepressant arm compared to controls was trended towards significant. (**C**) The antidepressant arm had a greater increase in task difficulty (a reduction in dot difference across stimuli) from baseline to follow-up, relative to controls. (**D**) Although not significant, the association between change in confidence and change in anxious-depression was in the expected negative direction in the antidepressant arm and among controls.

The online version of this article includes the following figure supplement(s) for figure 3:

**Figure supplement 1.** Changes in psychiatric dimensions and scale scores from baseline to follow-up in antidepressant (N=82) and control (N=88) arms using regression analyses.

dot difference was not significant (F(2, 1632)=1.91, p=0.148), suggesting no differences across the groups in task difficulty changes. In the antidepressant arm, mean dot difference decreased from baseline (M=41.2, SD=13.3) to follow-up (M=35.3, SD=13.1) ($\beta$=−5.91, SE = 1.25, p<0.001), indicating increased task difficulty. There was no significant change in task difficulty among controls from baseline (M=43.0, SD=11.8) to follow-up (M=41.4, SD=13.6) ($\beta$=−1.64, SE = 1.30, p=0.210) (*Figure 3C*). While our sample was underpowered to examine individual differences, we conducted an exploratory analysis examining the connection between changes in anxious-depression symptoms and changes in confidence in the antidepressant and controls groups. When examining the effects of time, group and anxious-depression change on mean confidence, there was a significant interaction effect of time and anxious-depression change on mean confidence (F(1, 1626)=4.04, p=0.045), suggesting change in confidence is associated with change in anxious-depression. There was no significant three-way interaction of anxious-depression change, time and group on mean confidence when comparing the three groups (F(2, 1626)=0.08, p=0.928), indicating that the significant association between confidence change and anxious-depression change was not specific to any group. Although not significant, the association between change in confidence and change in anxious-depression was in the expected negative direction in the antidepressant arm (r(80)=-0.10, p=0.381), and among controls (r(86)=-0.17, p=0.111) (*Figure 3D*).

## Discussion

Metacognitive biases are linked to transdiagnostic dimensions of mental health, but it is presently unclear if these biases are stable traits, or if they fluctuate alongside symptoms and change during the course of treatment (*Seow et al., 2021*). To answer these questions, we administered a previously validated adaptive task of metacognitive ability that controls for objective performance differences (*Fleming et al., 2014*) in a large sample of individuals before and after four weeks of iCBT or antidepressant medications (*Lee et al., 2023*). As expected, a 4-week course of iCBT or antidepressant medication led to transdiagnostic improvements in mental health (*Cipriani et al., 2018*; *Gøtzsche and Dinnage, 2020*; *Skapinakis et al., 2016*). Alongside this, there was a significant increase in metacognitive confidence following four weeks of iCBT or antidepressant medication. Not simply a practice effect, we found that individuals in the iCBT arm with the greatest improvements in anxious-depression had the largest increase in confidence at follow-up. This association with clinical improvements was specific to metacognitive changes, and not changes in task performance, suggesting that changes in confidence do not merely reflect greater task familiarity at follow-up. These findings suggest that metacognitive biases in anxious-depression are state-dependent. This builds on previous findings in small samples that have shown iCBT improves self-reported metacognitive self-beliefs (*Newby et al., 2014*) and that metacognition can be altered through adaptive training (*Carpenter et al., 2019*; *Engeler and Gilbert, 2020*; *Jelinek et al., 2017*; *Lysaker et al., 2018*). At baseline, we replicated the previously observed bi-directional associations between metacognitive bias and anxious-depression and compulsivity and intrusive thought (*Rouault et al., 2018*; *Seow and Gillan, 2020*; *Hoven et al., 2022*; *Benwell et al., 2022*). While higher levels of anxious-depression is associated with lower confidence, those with higher levels of compulsivity and intrusive thought have elevated confidence. This is a somewhat surprising dissociation, as compulsivity and anxious-depression are themselves positively correlated in the population. One way this can be reconciled is if the mechanisms underlying these opposing confidence biases are distinct. In anxious-depression, there appears to be more pervasive metacognitive biases that affect confidence in many domains and levels of a metacognitive hierarchy (spanning confidence in low level perceptual decisions to ideas of self-worth) (*Hoven et al., 2022*; *Seow et al., 2021*). In contrast, inflated confidence in compulsivity may be based on more specific biases in learning and inference (*Seow and Gillan, 2020*). The present study was observational and therefore did not randomly assign participants to a different treatments. To partially remediate that limitation, we included two smaller groups receiving antidepressant medication and a control group. Levels of transdiagnostic psychiatric dimensions remained stable across time among controls, while they significantly improved in the antidepressant arm. Similarly to iCBT, we found that confidence improved in the antidepressant group, but not among controls. The interaction, however, was not significant, meaning that we cannot reject the null hypothesis that confidence improved to the same degree across the three groups. As increased task difficulty among clinical groups was not significantly greater relative to controls, changes in task difficulty may simply reflect greater task familiarity at

follow-up across groups, as opposed to gains in general perceptual performance among clinical arms. Examining the three groups together, the data suggests that confidence changes are unlikely to be treatment specific, rather, confidence fluctuates in tandem with anxious-depression. This was evident in an overall association between change in anxious-depression and change in confidence that was not modified by treatment arm. Additionally, levels of iCBT engagement and concurrent treatments did not bolster changes in confidence. Overall, the results indicated that metacognition fluctuates with anxious-depression state, regardless of treatment type or exposure. Future research with larger samples are required to address this definitively.

## Limitations and future directions

Confidence change and anxious-depression change were significantly but weakly associated. Similarly, the relative change in confidence across treatment arms was small. Therefore, while tests of metacognitive confidence can inform theoretical models, like most cognitive tests, they are likely of limited utility in clinical practice, at least when used in isolation (*Mogoașe et al., 2017*). Given the complexity of mental health causes and presentations, multivariable models are needed to see practical value from such tests. We did not assess confidence or anxious-depression to treatment cessation and so the causal path and temporal dependence, if they exist, cannot be derived from these data. Future research should consider assessing metacognition and anxious-depression continuously through treatment, in order to elucidate the causal relationship between anxious-depression and metacognition with mediation analysis (*Newby et al., 2014*). More intensive, repeating testing in future studies may also reveal the temporal window at which metacognition has the propensity to change, which could be more momentary in nature. While this study examined changes in metacognition with iCBT generally, future research should examine if the strength of the association between confidence change and anxious-depression change is greater following iCBT modules targeting metacognition or following metacognitive intervention (*Philipp et al., 2020*). The iCBT programs in this study primarily targeted depression and anxiety, which may explain why changes in confidence did not scale with improvements in compulsivity. Future research is required to assess if treatments aimed at compulsive disorders decrease the over-confidence commonly observed in those high in compulsivity and intrusive thought. As the antidepressant and control groups were much smaller than the iCBT arm, we were unable to compare changes in confidence across the types of antidepressant medications individuals received and we were underpowered more generally for individual differences analyses and multi-arm comparisons. Exploratory analyses were nonetheless presented and can form the basis for future investigations.

## Conclusions

Our findings replicated the cross-sectional evidence that higher levels of anxious-depression are associated with under-confidence. We demonstrate that metacognitive confidence increases following four weeks of iCBT or antidepressant treatment. Overall, we observed that the greater the improvement in anxious-depression, the more confident participants became, which did not appear to be dependent on treatment type. This suggests that metacognitive biases in anxious-depression are state-dependent and might be normalised through clinical improvements.

## Methods

### Participants

Participants were recruited as part of the Precision in Psychiatry (PIP) study (*Lee et al., 2023*), an observational, longitudinal study in which participants underwent a 4-week course of iCBT or antidepressant medication. Further details of the PIP study procedures that are not specific to this study can be found in a prior publication (*Lee et al., 2023*). Ethical approval for the PIP study was obtained from the Research Ethics Committee of School of Psychology, Trinity College Dublin (Approval ID: SPREC072017-01) and the Northwest-Greater Manchester West Research Ethics Committee of the National Health Service, Health Research Authority and Health and Care Research Wales (REC reference: 20/NW/0020, IRAS project ID: 270623). Informed consent, and consent to publish, was obtained from all participants prior to study participation. A power analysis was carried out using effect sizes from a previous study examining cross-sectional associations between metacognition and

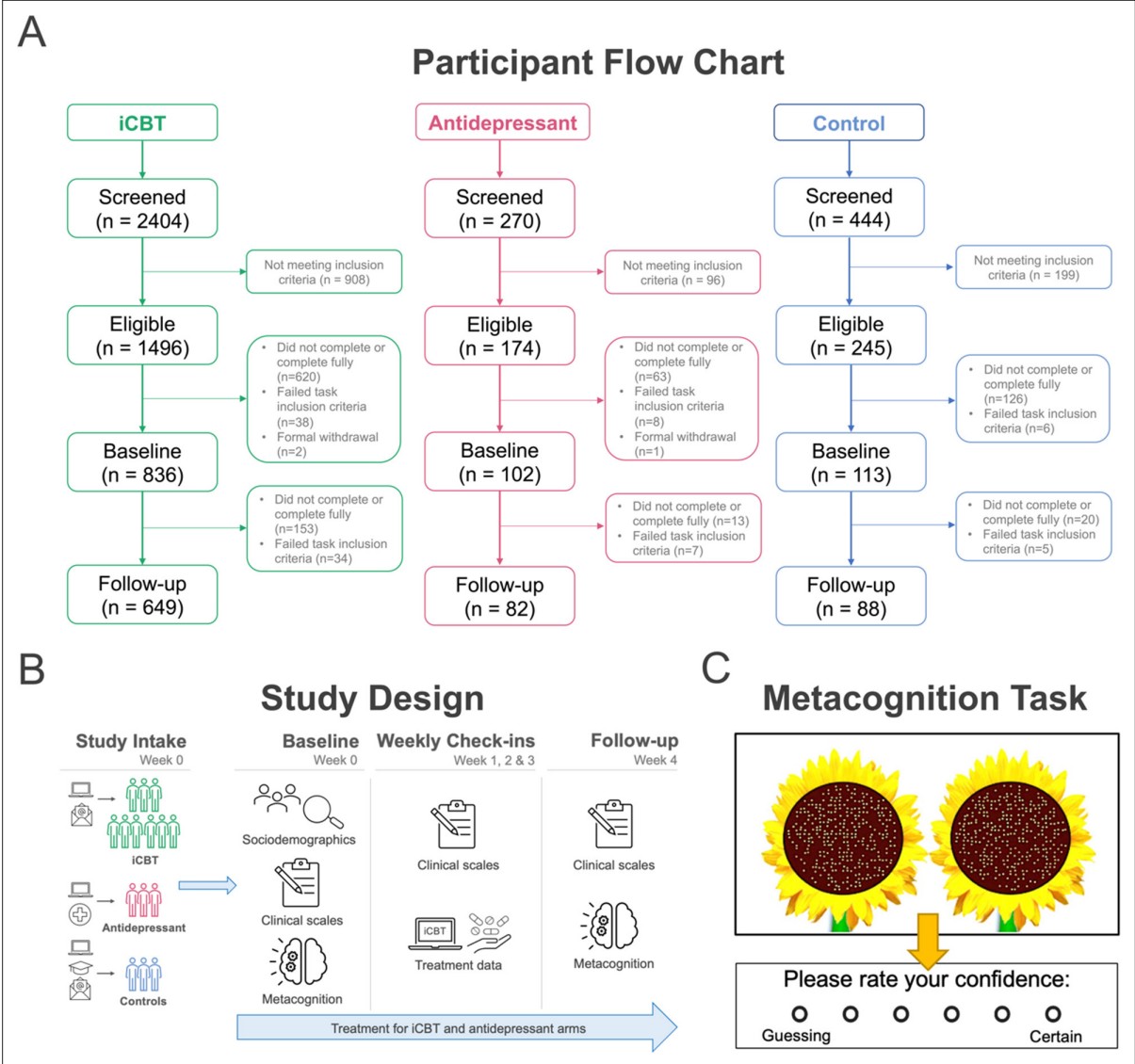

**Figure 4.** Study methods. (**A**) Participant flow chart (CONSORT chart). Participants were considered 'completers' if they had metacognitive and transdiagnostic psychiatric dimension data at baseline and follow-up and met task inclusion criteria. (**B**) Overview of study design from study intake (week 0) to follow-up (week 4) assessments across groups. (**C**) Metacognitive (visuo-perceptual decision-making) task design (N=210 trials). On each trial, participants were asked to judge and choose the sunflower that contained more seeds (i.e. higher number of dots) and then provide a confidence rating on their decision.

anxious-depression, and compulsivity and intrusive thought (*Rouault et al., 2018*). Sample sizes of N=454 and N=332 respectively were required to detect these associations with 80% power. The sample sizes of the antidepressant arm and control group were smaller and used for secondary and more exploratory analyses.

## iCBT Arm

Individuals initiating iCBT provided by SilverCloud Health were recruited from two sites: (1) the National Health Service Berkshire Foundation in the UK and (2) Aware mental health charity in Ireland. Participants included in the study either started their iCBT intervention ≤2 days prior to signing up, or provided a treatment start date in the near future, and scored ≥10 on the Work and Social Adjustment Scale (WSAS) at baseline, which indicated significant functional impairment due to clinical symptoms (*Mundt et al., 2002*). *Figure 4A* shows the disposition of participants throughout the study. N=2404 were screened, of whom N=1,496 were eligible, N=836 completed baseline assessments and met

**Table 1.** Baseline sociodemographic characteristics of participants.

| Characteristic | iCBT (n=649)* | Antidepressant (n=82) | Control (n=88) | F/X² (df) | p |
|---|---|---|---|---|---|
| Gender, No. (%) | | | | | 0.414[†] |
| Male | 141 (21.8) | 20 (24.4) | 22 (25.0) | | |
| Female | 501 (77.4) | 60 (73.2) | 66 (75.0) | | |
| Other | 5 (0.8) | 2 (2.4) | 0 (0.0) | | |
| Age, M (SD) | 32.2 (11.0) | 30.5 (10.5) | 29.1 (12.0) | 3.68 (2, 813) | 0.026 |
| Country of residence, No. (%) | | | | 211.73 (4) | <0.001 |
| Ireland | 89 (13.8) | 32 (39.0) | 58 (65.9) | | |
| United Kingdom | 546 (84.4) | 34 (41.5) | 24 (27.3) | | |
| Other | 12 (1.9) | 16 (19.5) | 6 (6.8) | | |
| Highest level of education, No. (%) | | | | 7.11 (4) | 0.130 |
| Below undergraduate | 147 (22.7) | 12 (14.6) | 13 (14.8) | | |
| Some/completed undergraduate | 342 (52.9) | 49 (59.8) | 57 (64.8) | | |
| Above undergraduate | 158 (24.4) | 21 (25.6) | 18 (20.5) | | |

*Two participants in the iCBT arm were missing data for age, gender, country of residence and highest level of education.

[†]Gender proportions were compared using Fisher's exact due to cell count <5 in the 'Other' group.

inclusion criteria. A final N=649 completed and met inclusion criteria for follow-up assessments. While study follow-up data was collected after four weeks of treatment, iCBT could last up to 12 weeks (*Lee et al., 2023*). The final sample was, on average, 32.2 years old (SD=11.0), mostly female (n=501, 77.4%), living in the United Kingdom (n=546, 84.4%), and had some or completed undergraduate level education (n=342, 52.9%) (*Table 1*).

## Antidepressant Arm

Individuals were recruited globally using advertisements placed on Google search, in addition to social media platforms, mental health websites, local pharmacies and General Practitioner waiting rooms. Participants were included if they started or planned to start treatment ≤2 days of study sign-up, scored ≥10 on the WSAS at baseline, and provided a valid photograph of an antidepressant medication prescription. N=270 individuals were screened, of whom N=174 were eligible, N=102 completed and met inclusion criteria at baseline and a final N=82 had follow-up data (*Figure 4A*). Participants were mostly female (n=60, 73.2%), mean age = 30.5 (10.5), were living in Ireland or the United Kingdom (n=66, 80.5%) and had some or completed undergraduate level education (n=49, 59.8%) (*Table 1*).

## Control group

Participants in the no treatment control group were recruited through university mailings lists and advertisements posted online and around Trinity College Dublin. Participants included in this arm scored <10 on an adapted version of the WSAS (where they rated functional impairment from their general problems rather than mental health problems) and self-reported that they had no current mental health problems and were not undergoing treatment for any mental health problems at the time of screening. N=444 individuals were screened, of whom N=245 were eligible, N=113 had baseline data and a final N=88 completed follow-up assessments and met inclusion criteria for the study (*Figure 4A*). Participants in the control group were matched for sociodemographic characteristics in the antidepressant arm, with no significant differences across the two groups in gender (Fisher's exact test p=0.498), or age (Welch's t(166.66)=−0.81, p=0.420), or levels of educational attainment

($\chi^2$(2)=0.66, p=0.718). Participants in the control group came from the United Kingdom and Ireland (>93%), whereas the antidepressant arm was more international, with 20% coming from other countries ($\chi^2$(2)=13.59, p=0.001).

When comparing sociodemographic characteristics across the three study groups, there were no significant differences between groups in gender proportions (Fisher's exact test p=0.414), or levels of educational attainment ($\chi^2$(4)=7.11, p=0.130), as reported in *Table 1*. Age differed across the three arms (F(2, 813)=3.68, p=0.026), with post hoc Tukey tests indicating that mean age was higher in the iCBT arm (M=32.2, SD=11.0) compared to the controls (M=29.1, SD=12.0) (p$_{adj}$ = 0.034), but not when compared to the antidepressant arm (M=30.5, SD=10.5) (p$_{adj}$ = 0.376) (*Table 1*). The countries participants were living in varied across the study arms ($\chi^2$(4)=211.73, p<0.001), as there was a higher proportion of individuals in the iCBT arm living in the UK (n=546, 84.4%) when compared to the antidepressant arm (n=34, 41.5%) and the control group (n=24, 27.3%) (*Table 1*).

## Procedure

*Figure 4B* shows an overview of the study design, including the assessments involved at each timepoint. For the purposes of this study, we focused on a select set of sociodemographic characteristics (gender, age, country of residence, level of educational attainment), self-reported psychiatric questionnaires, metacognitive task performance and treatment data from the PIP study (*Lee et al., 2023*).

## Self-reported psychiatric questionnaires

Participants completed the WSAS (*Mundt et al., 2002*) at baseline and follow-up. Individuals in the iCBT and antidepressant arms received the original WSAS, which asks them to rate their level of functional impairment from mental health problems. Individuals in the control group received an adapted version of the WSAS, which asks them to rate their level of impairment from 'their problems' more generally. Each WSAS item was scored from 0 'not at all' to 8 'very severely', with overall scores ranging from 0 to 40. Higher WSAS scores indicated higher levels of functional impairment (*Mundt et al., 2002*). Participants completed nine standard self-report clinical questionnaires at baseline and four-week follow-up that assess a variety of psychiatric symptoms, including depression (Zung Self-Rating Depression Scale; *Zung, 1965*), trait anxiety (State Trait Anxiety Inventory) (*Spielberger et al., 1983*), schizotypy (Short Scales for Measuring Schizotypy) (*Mason et al., 2005*), impulsivity (Barratt Impulsiveness Scale 11) (*Patton et al., 1995*), obsessive-compulsive disorder (OCD) (Obsessive-Compulsive Inventory-Revised, OCI-R) (*Foa et al., 2002*), social anxiety (Liebowitz Social Anxiety Scale; *Liebowitz, 1987*), eating disorders (Eating Attitudes Test) (*Garner et al., 1982*), apathy (Apathy Evaluation Scale) (*Marin et al., 1991*), and alcohol misuse (Alcohol Use Disorders Identification Test) (*Saunders et al., 1993*).

## Metacognitive task

Participants completed a visuo-perceptual decision-making task (*Fleming et al., 2014*) to assess metacognition (*Figure 4C*). On each trial, participants were shown a fixation cross for 1000 milliseconds (ms), followed by two sunflowers with a varying number of seeds for 300ms. Participants then had unlimited time to make a judgement on which of two sunflower stimuli contained more seeds and then rated their confidence in each judgement, on a scale from '1=Guessing' to '6=Certain'. There was a total of 210 trials, divided equally into five blocks. While participants were given feedback on which sunflower they had chosen for 500ms, there was no trial-level feedback provided on their performance. Split-half reliability was high for baseline mean confidence, as indicated by the correlation between odd and even trials across each participant in the iCBT arm (r(647)=0.98, p<0.001). Mean accuracy was tightly controlled in this task using a 'two-down one-up' staircase procedure, in which equally sized changes in dot difference occurred after each incorrect response and after two consecutive correct responses. This maintained objective performance across all participants at a desired level of 70% correct, crucial for estimating metacognition without any confound of real performance differences. Changes in dot difference were calculated using log-space, with a start log difference of 4.2 (+70 dots). Differences in step size changed by ± 0.4 for the first five trials, ± 0.2 for the next five trials and ± 0.1 for the remainder of the task. Dot difference on each trial could range from six dots (1.79 in log-space, the most difficult to discriminate) to 81 dots (4.39 in log-space, the easiest to discriminate).

## Treatment data

Every week, participants self-reported if they had adhered to iCBT or their antidepressant medications each week. Participants also reported if they had started any new concurrent medications and/or psychological treatments for mental health each week. Starting new medication and/or psychological treatments and treatment nonadherence during the study were not grounds for exclusion in any group. Objective indicators of treatment engagement were provided by SilverCloud for 640 participants in the iCBT arm, which comprised of percentage of program viewed, time (minutes) spent in the program, and program type. In relation to program type, *Space from Depression* was the most common iCBT program (n=158, 24.7%), followed by *Space from Generalised Anxiety Disorder* (n=96, 15.0%), *Life Skills Online* (n=89, 13.9%), *Space from Anxiety* (n=84, 13.1%), and *Space from Depression and Anxiety* (n=77, 12.0%). Given the overlapping content of *Space from Generalised Anxiety Disorder* and *Space from Anxiety*, these programs were merged into one category '*Space from Anxiety*' (n=180, 28.1%) for analyses. The '*Other*' category (n=136, 21.3%) comprised of the following iCBT programs: *Space from Stress* (n=33, 5.2%), *Space from Social Anxiety* (n=20, 3.1%), *Space for Resilience* (n=20, 3.1%), *Space for Perinatal Wellbeing* (n=18, 2.8%), *Space in Chronic Pain from Depression and Anxiety* (n=12, 1.9%), *Space from Health Anxiety* (n=9, 1.4%), *Space from OCD* (n=7, 1.1%), *Space from Panic* (n=6, 0.9%), *Space from Phobia* (n=5, 0.8%), *Space for Sleep* (n=4, 0.6%), *Space in Lung Conditions from Depression and Anxiety* (n=1, 0.2%) and *Space from Money Worries* (n=1, 0.2%), which were merged together into one category for analysis due to low N in each program.

## Data preparation and analysis

### Self-reported psychiatric questionnaire

Individual scores on dimensions of anxious-depression, compulsivity and intrusive thought, and social withdrawal were calculated by multiplying each of the 209 item scores on the nine above-mentioned self-report clinical questionnaires by the 209 corresponding item weights from a previously published factor analysis on these scales (*Gillan et al., 2016*). Dimension scores were scaled to centre on zero, with higher scores indicating higher levels of transdiagnostic psychopathology.

To determine the proportion of careless/inattentive responders on the self-report clinical questionnaire, we included 'catch' questions that were embedded in the OCI-R (*If you are paying attention to these questions, please select "A little"*) and the WSAS (*If you are paying attention to these questions, please select "Not at all"*) at baseline and in the WSAS at all subsequent assessments. In the iCBT arm, 54 (8.3%) participants failed at least one of the checks, with just nine (1.4%) failing more than one. Additionally, eight (9.8%) individuals in the antidepressant arm and seven (8.0%) in the control group failed at least one of the catch items. Given the small number of participants that failed more than one attention checks and that all those participants passed the task exclusion criteria, these individuals were retained for subsequent analyses. Additionally, excluding those that failed more than one catch item in the iCBT arm did not affect the significance of results, including the change in confidence ($\beta$=0.16, SE = 0.02, p<0.001), change in anxious-depression ($\beta$=−0.32, SE = 0.03, p<0.001), and the association between change in confidence and change in anxious-depression (r(638)=-0.10, p=0.011).

### Metacognition task

The perceptual decision-making task performance was used to quantify our primary cognitive outcome measure, metacognitive bias, the mean confidence rating across trials. Task difficulty was measured as the mean dot difference across trials, where more difficult trials had a lower dot difference between stimuli. Mean reaction time to stimuli choice across trials was measured in seconds and task accuracy was calculated as the mean proportion of correct responses across trials. As metacognitive efficiency (the extent to which confidence estimates map onto objective performance) was not previously associated with transdiagnostic dimensions cross-sectionally (*Rouault et al., 2018*; *Hoven et al., 2022*; *Benwell et al., 2022*), the state-dependence of efficiency was not examined in this study.

We employed a number of exclusion criteria to ensure high data quality from the metacognitive task. Firstly, we planned to exclude participants if they selected the right or left sunflower on greater than 95% of trials, but none met this criterion. Second, we excluded participants with mean accuracy less than 0.60 and greater than 0.85, indicating the staircase procedure did not converge within acceptable bounds (n=38 at baseline and n=34 at follow-up excluded in the iCBT arm; n=8

at baseline and n=7 at follow-up excluded in the antidepressant arm; n=6 at baseline and n=5 at follow-up excluded in the control group).

## Treatment data

Treatment adherence was high by week 3 in both clinical arms, with over ≥95% of the iCBT arm still undergoing treatment (i.e. 98% at weekly check-in (WCI) 1, 97% at WCI 2 and 95% at WCI 3) and ≥99% of the antidepressant group reported still taking antidepressant medication (i.e. 100% at WCI 1 and WCI 2 and 99% at WCI 3). In the antidepressant arm, four participants altered the dosage of their medication during the study participation (n=2 took less than prescribed and n=2 took more than prescribed). In terms of concurrent treatment, 175 (27.0%) in the iCBT arm were receiving another treatment during the study, of which 48 (7.4%) were taking concurrent medication for a mental health problem and 145 (22.3%) were receiving a concurrent form of psychotherapy. For the antidepressant group, 33 (40.2%) were receiving another treatment during the study, with 6 (7.3%) taking at least one other medication for a mental health problem and 2 (39.0%) were receiving some form of psychotherapy. Thus, there were partial overlaps in the treatments received across the two clinical arms. Within the control group, two participants (2.3%) started taking medication during the study and none initiated psychotherapy.

## Statistical analysis

We tested for relationships between baseline task measures and the psychiatric symptom dimensions using linear regression analysis, controlling for age, gender, and education. To examine pre-post changes, we carried out linear mixed-effects models with measures of metacognition or psychopathology as the dependent variable, time (baseline = 0; follow-up=1) as the independent variable and participants as random effects. To determine the association between change in confidence and change in anxious-depression, we used (1) Pearson correlation analysis to correlate change indices for both measures and, (2) regression-based repeated-measures analysis: mean confidence ~time*anxious-depression score change, where mean confidence is entered with two datapoints (one for pre- and one for post-treatment i.e., within-person) and anxious-depression change is a single value per person (between-person change score). Exploratory linear regression analyses tested the specificity of the effects, replacing anxious-depression with each of the measures of psychopathology in turn as follows: Mean confidence ~time*dimension/psychiatric scale score change. We additionally ran regression analyses to test if concurrent treatment or the degree of objective engagement in iCBT interacted with the effect of time on mean confidence. Exploratory ANOVA analyses were also conducted to compare changes in anxious-depression, task difficulty and confidence across the three arms directly. For all tests, statistical significance was defined as $p < 0.05$, with two-tailed p-values used. All regressors were scaled as Z scores to compare the regression coefficients of independent variables within each model. Adjustments for multiple comparisons were not conducted for analyses of replicated effects, or exploratory analyses (*Althouse, 2016*). The code to reproduce statistical analyses are available at https://osf.io/89xzq/. The datasets used and/or analysed during the current study are available from the corresponding author on reasonable request. All data from the PIP study will be made publicly available once the primary treatment prediction analysis of the project are published on a pre-print server.

## Acknowledgements

This work was funded by a fellowship awarded to Claire M Gillan from MQ: transforming mental health (MQ16IP13). Claire M Gillan holds additional funding from Science Foundation Ireland's Frontiers for the Future Scheme (19/FFP/6418), and a European Research Council (ERC) Starting Grant (ERC-H2020-HABIT). The PhD studentship of Celine Ann Fox is funded by the Government of Ireland Postgraduate Scholarship Programme (GOIPG/2020/662). The authors thank all the participants for their involvement in this study. We thank the AWARE charity and the Berkshire foundation trust that supported recruitment for the iCBT arm. We thank the individual pharmacies and General Practitioner services for their support in recruiting the antidepressant arm.

# Additional information

## Competing interests

Chi Tak Lee: The PhD studentship of Chi Tak Lee is co-funded by SilverCloud Health and the Irish Research Council. Siobhán Harty: Siobhán Harty is a current employees of SilverCloud Health. Derek Richards: Derek Richards is a current employees of SilverCloud Health. Jorge Palacios: Jorge Palacios is a current employees of SilverCloud Health. Klaas Enno Stephan: Klaas Enno Stephan acknowledges support by the René and Susanne Braginsky Foundation and the ETH Foundation. The other authors declare that no competing interests exist.

## Funding

| Funder | Grant reference number | Author |
| --- | --- | --- |
| MQ: Transforming Mental Health | MQ16IP13 | Claire M Gillan |
| Science Foundation Ireland | 19/FFP/6418 | Claire M Gillan |
| European Research Council | ERC-H2020-HABIT | Claire M Gillan |
| Government of Ireland Postgraduate Scholarship Programme | GOIPG/2020/662 | Celine Ann Fox |
| Irish Research Council | Enterprise Partnership Scheme IRC/EPSPG/2020/8 | Chi Tak Lee Claire M Gillan |

The funders had no role in study design, data collection and interpretation, or the decision to submit the work for publication.

## Author contributions

Celine Ann Fox, Conceptualization, Formal analysis, Funding acquisition, Investigation, Visualization, Writing – original draft, Writing – review and editing; Chi Tak Lee, Data curation, Formal analysis, Investigation, Writing – review and editing; Anna Kathleen Hanlon, Kevin Lynch, Investigation, Project administration, Writing – review and editing; Tricia XF Seow, Resources, Methodology; Siobhán Harty, Derek Richards, Jorge Palacios, Veronica O'Keane, Klaas Enno Stephan, Methodology, Writing – review and editing; Claire M Gillan, Funding acquisition, Methodology, Writing – original draft, Writing – review and editing

## Author ORCIDs

Celine Ann Fox ⓘ https://orcid.org/0000-0003-1740-3765
Tricia XF Seow ⓘ https://orcid.org/0000-0002-5930-8929
Derek Richards ⓘ https://orcid.org/0000-0003-0871-4078
Klaas Enno Stephan ⓘ https://orcid.org/0000-0002-8594-9092
Claire M Gillan ⓘ http://orcid.org/0000-0001-9065-403X

## Ethics

Ethical approval for the PIP study was obtained from the Research Ethics Committee of School of Psychology, Trinity College Dublin (Approval ID: SPREC072017-01) and the Northwest-Greater Manchester West Research Ethics Committee of the National Health Service, Health Research Authority and Health and Care Research Wales (REC reference: 20/NW/0020, IRAS project ID: 270623). Informed consent, and consent to publish, was obtained from all participants prior to study participation.

Reviewer #1 (Public Review): https://doi.org/10.7554/eLife.87193.3.sa1
Reviewer #2 (Public Review): https://doi.org/10.7554/eLife.87193.3.sa2
Reviewer #3 (Public Review): https://doi.org/10.7554/eLife.87193.3.sa3
Author Response: https://doi.org/10.7554/eLife.87193.3.sa4

## Additional files

### Supplementary files
• MDAR checklist

### Data availability
The code to reproduce statistical analyses are available at https://osf.io/89xzq/. The datasets used and/or analysed during the current study are available from the corresponding author on reasonable request. All data from the PIP study will be made publicly available once the primary treatment prediction analysis of the project are published.

The following dataset was generated:

| Author(s) | Year | Dataset title | Dataset URL | Database and Identifier |
|-----------|------|---------------|-------------|-------------------------|
| Fox CA, Gillan C | 2023 | An observational treatment study of metacognition in anxious-depression | https://osf.io/89xzq/ | Open Science Framework, 89xzq |

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
