## [Editor Report · eLife assessment]

This **valuable** study advances our knowledge of the effects of anxiety/depression treatment on metacognition, demonstrating that treatment increases metacognitive confidence alongside improving symptoms. The authors provide **convincing** evidence for the state-dependency of metacognitive confidence, based on a large longitudinal treatment dataset. However, it is unclear to what extent this effect is truly specific to treatment, as changes in metacognitive confidence in the group receiving online therapy were not statistically different from those in the control group.

---

## [Referee Report · Reviewer #1 (Public Review)]

It has been shown previously that there are relationships between a transdiagnostic construct of anxious-depression (AD), and average confidence rating in a perceptual decision task. This study sought to investigate these results, which have been replicated several times but only in cross-sectional studies. This work applies a perceptual decision-making task with confidence ratings and a transdiagnostic psychometric questionnaire battery to participants before and after an iCBT course. The iCBT course reduced AD scores in participants, and their mean confidence ratings increased without a change in performance. Participants with larger AD changes had larger confidence changes. These results were also shown in a separate smaller group receiving antidepressant medication. A similar sized control group with no intervention did not show changes.

The major strength of the study is the elegant and well-powered data set. Longitudinal data on this scale is very difficult to collect, especially with patient cohorts, so this approach represents an exciting breakthrough. Analysis is straightforward and clearly presented. However, no multiple comparison correction is applied despite many different tests. While in general I am not convinced of the argument in the citation provided to justify this, I think in this case the key results are not borderline (p<0.001) and many of the key effects are replications, so there are not so many novel/exploratory hypothesis and in my opinion the results are convincing and robust as they are. The supplemental material is a comprehensive description of the data set, which is a useful resource.

The authors achieved their aims, and the results clearly support the conclusion that the AD and mean confidence in a perceptual task covary longitudinally.

I think this study provides an important impact to the project of computational psychiatry.Sspecifically, it shows that the relationship between transdiagnostic symptom dimensions and behaviour is meaningful within as well as across individuals.

---

## [Referee Report · Reviewer #2 (Public Review)]

The authors of this study investigated the relationship between (under)confidence and the anxious-depressive symptom dimension in a longitudinal intervention design. The aim was to determine whether confidence bias improves in a state-like manner when symptoms improve. The primary focus was on patients receiving internet-based CBT (iCBT; n=649), while secondary aims compared these changes to patients receiving antidepressants (n=82) and a control group (n=88).

The results support the authors' conclusions, and the authors convincingly demonstrated a weak link between changes in confidence bias and anxious-depressive symptoms (not specific to the intervention arm)

The major strength and contribution of this study is the use of a longitudinal intervention design, allowing the investigation of how the well-established link between underconfidence and anxious-depressive symptoms changes after treatment. Furthermore, the large sample size of the iCBT group is commendable. The authors employed well-established measures of metacognition and clinical symptoms, used appropriate analyses, and thoroughly examined the specificity of the observed effects.

However, due to the small effect sizes, the antidepressant and control groups were underpowered, reducing comparability between interventions and the generalizability of the results. The lack of interaction effect with treatment makes it harder to interpret the observed differences in confidence, and practice effects could conceivably account for part of the difference. Finally, it was not completely clear to me why, in the exploratory analyses, the authors looked at the interaction of time and symptom change (and group), since time is already included in the symptom change index.

This longitudinal study informs the field of metacognition in mental health about the changeability of biases in confidence. It advances our understanding of the link between anxiety-depression and underconfidence consistently found in cross-sectional studies. The small effects, however, call the clinical relevance of the findings into question. I would have found it useful to read more in the discussion about the implications of the findings (e.g., why is it important to know that the confidence bias is state-dependent; given the effect size of the association between changes in confidence and symptoms, is the state-trait dichotomy the right framework for interpreting these results; suggestions for follow-up studies to better understand the association).

---

## [Referee Report · Reviewer #3 (Public Review)]

This study reports data collected across time and treatment modalities (internet CBT (iCBT), pharmacological intervention, and control), with a particularly large sample in the iCBT group. This study addresses the question of whether metacognitive confidence is related to mental health symptoms in a trait-like manner, or whether it shows state-dependency. The authors report an increase in metacognitive confidence as anxious-depression symptoms improve with iCBT (and the extent to which confidence increases is related to the magnitude of symptom improvement), a finding that is largely mirrored in those who receive antidepressants (without the correlation between symptom change and confidence change). I think these findings are exciting because they directly relate to one of the big assumptions when relating cognition to mental health - are we measuring something that changes with treatment (is malleable), so might be mechanistically relevant, or even useful as a biomarker?

This work is also useful in that it replicates a finding of heightened confidence in those with compulsivity, and lowered confidence in those with elevated anxious-depression.

One caveat to the interest of this work is that it doesn't allow any causal conclusions to be drawn, and only measures two timepoints, so it's hard to tell if changes in confidence might drive treatment effects (but this would be another study). The authors do mention this in the limitations section of the paper.

Another caveat is the small sample in the antidepressant group.

Some thoughts I had whilst reading this paper: to what extent should we be confident that the changes are not purely due to practice? I appreciate there is a relationship between improvement in symptoms and confidence in the iCBT group, but this doesn't completely rule out a practice effect (for instance, you can imagine a scenario in which those whose symptoms have improved are more likely to benefit from previously having practiced the task).

Relatedly, to what extent is there a role for general task engagement in these findings? The paper might be strengthened by some kind of control analysis, perhaps using (as a proxy for engagement) the data collected about those who missed catch questions in the questionnaires.

I was also unclear what the findings about task difficulty might mean. Are confidence changes purely secondary to improvements in task performance generally - so confidence might not actually be 'interesting' as a construct in itself? The authors could have commented more on this issue in the discussion.

To make code more reproducible, the authors could have produced an R notebook that could be opened in the browser without someone downloading the data, so they could get a sense of the analyses without fully reproducing them.

Rather than reporting full study details in another publication I would have found it useful if all relevant information was included in a supplement (though it seems much of it is). This avoids situations where the other publication is inaccessible (due to different access regimes) and minimises barriers for people to fully understand the reported data.

---

## [Author Response]

**Reviewer #1 (Public Review):**
[…] The major strength of the study is the elegant and well-powered data set. Longitudinal data on this scale is very difficult to collect, especially with patient cohorts, so this approach represents an exciting breakthrough. Analysis is straightforward and clearly presented. However, no multiple comparison correction is applied despite many different tests. While in general I am not convinced of the argument in the citation provided to justify this, I think in this case the key results are not borderline (p<0.001) and many of the key effects are replications, so there are not so many novel/exploratory hypothesis and in my opinion the results are convincing and robust as they are. The supplemental material is a comprehensive description of the data set, which is a useful resource.The authors achieved their aims, and the results clearly support the conclusion that the AD and mean confidence in a perceptual task covary longitudinally. I think this study provides an important impact to the project of computational psychiatry.Sspecifically, it shows that the relationship between transdiagnostic symptom dimensions and behaviour is meaningful within as well as across individuals.

Response: We thank the reviewer for their appraisal of our paper and positive feedback on the main manuscript and supplementary information. We agree with the reviewer that the lack of multiple comparison corrections can also justified by key findings being replications and not borderline significance. We have added this additional justification to the manuscript (Methods, Statistical Analyses, page 15, line 568: “Adjustments for multiple comparisons were not conducted for analyses of replicated effects”)

**Reviewer #2 (Public Review):**
[…] The major strength and contribution of this study is the use of a longitudinal intervention design, allowing the investigation of how the well-established link between underconfidence and anxious-depressive symptoms changes after treatment. Furthermore, the large sample size of the iCBT group is commendable. The authors employed well-established measures of metacognition and clinical symptoms, used appropriate analyses, and thoroughly examined the specificity of the observed effects.However, due to the small effect sizes, the antidepressant and control groups were underpowered, reducing comparability between interventions and the generalizability of the results. The lack of interaction effect with treatment makes it harder to interpret the observed differences in confidence, and practice effects could conceivably account for part of the difference. Finally, it was not completely clear to me why, in the exploratory analyses, the authors looked at the interaction of time and symptom change (and group), since time is already included in the symptom change index.

Response: We thank the reviewer for their succinct summary of the main results and strengths of our study. We apologise for the confusion in how we described that analysis. We examine state-dependence., i.e. the relationship between symptom change and metacognition change, in two ways in the paper – perhaps somewhat redundantly. (1) By correlating change indices for both measures (e.g. as plotted in Figure 3D) and (2) by doing a very similar regression-based repeated-measures analysis, i.e. mean confidence ~ time*anxious-depression score change. Where mean confidence is entered with two datapoints – one for pre- and one for post-treatment (i.e. within-person) and anxious-depression change is a single value per person (between-person change score). This allowed us to test if those with the biggest change in depression had a larger effect of time on confidence. This has been added to the paper for clarification (Methods, Statistical Analysis, page 14, line 553-559: “To determine the association between change in confidence and change in anxious-depression, we used (1) Pearson correlation analysis to correlate change indices for both measures and, (2) regression-based repeated-measures analysis: mean confidence ~ time*anxious-depression score change, where mean confidence is entered with two datapoints (one for pre- and one for post-treatment i.e., within-person) and anxious-depression change is a single value per person (between-person change score)”).

The analyses have also been reported as regression in the Results for consistency (Treatment Findings: iCBT, page 5, line 197-204: ‘To test if changes in confidence from baseline to follow-up scaled with changes in anxious-depression, we ran a repeated measure regression analyses with per-person changes in anxious-depression as an additional independent variable**.** We found this was the case, evidenced by a significant interaction effect of time and change in anxious-depression on confidence (b=-0.12, SE=0.04, p=0.002)… This was similarly evident in a simple correlation between change in confidence and change in anxious-depression (r(647)=-0.12, p=0.002)”).

This longitudinal study informs the field of metacognition in mental health about the changeability of biases in confidence. It advances our understanding of the link between anxiety-depression and underconfidence consistently found in cross-sectional studies. The small effects, however, call the clinical relevance of the findings into question. I would have found it useful to read more in the discussion about the implications of the findings (e.g., why is it important to know that the confidence bias is state-dependent; given the effect size of the association between changes in confidence and symptoms, is the state-trait dichotomy the right framework for interpreting these results; suggestions for follow-up studies to better understand the association).

Response: Thank you for this comment. We have elaborated on the implications of our findings in the Discussion, including the relevance of the state-trait dichotomy to future research and how more intensive, repeated testing may inform our understanding of the state-like nature of metacognition (Discussion, Limitations and Future Directions, page 10, line 378-380: “More intensive, repeating testing in future studies may also reveal the temporal window at which metacognition has the propensity to change, which could be more momentary in nature.”).

**Reviewer #3 (Public Review):**
[…] I think these findings are exciting because they directly relate to one of the big assumptions when relating cognition to mental health - are we measuring something that changes with treatment (is malleable), so might be mechanistically relevant, or even useful as a biomarker?This work is also useful in that it replicates a finding of heightened confidence in those with compulsivity, and lowered confidence in those with elevated anxious-depression.One caveat to the interest of this work is that it doesn't allow any causal conclusions to be drawn, and only measures two timepoints, so it's hard to tell if changes in confidence might drive treatment effects (but this would be another study). The authors do mention this in the limitations section of the paper.Another caveat is the small sample in the antidepressant group.Some thoughts I had whilst reading this paper: to what extent should we be confident that the changes are not purely due to practice? I appreciate there is a relationship between improvement in symptoms and confidence in the iCBT group, but this doesn't completely rule out a practice effect (for instance, you can imagine a scenario in which those whose symptoms have improved are more likely to benefit from previously having practiced the task).

Response: We thank the reviewer for commenting on the implications of our findings and we agree with the caveats listed. We thank the reviewer for raising this point about practice effects. A key thing to note is that this task does not have a learning element with respect to the core perceptual judgement (i.e., accuracy), which is the target of the confidence judgment itself. While there is a possibility of increased familiarity with the task instructions and procedures with repeated testing, the task is designed to adjust the difficulty to account of any improvements, so accuracy is stable. We see that we may not have made this clear in some of our language around accuracy vs. perceptual difficulty and have edited the Results to make this distinction clearer (Treatment Findings: iCBT, pages 4-5, lines 184-189: “Although overall accuracy remained stable due to the staircasing procedure, participants’ ability to detect differences between the visual stimuli improved. This was reflected as the overall increase in task difficulty to maintain the accuracy rates from baseline (dot difference: M=41.82, SD=11.61) to follow-up (dot difference: M=39.80, SD=12.62), (b=-2.02, SE=0.44, p<0.001, r2=0.01)”.)

However, it is true that there can be a ‘practice’ effect in the sense that one may feel more confident (despite the same accuracy level) due to familiarity with a task. One reason we do not subscribe to the proposed explanation for the link between anxious-depression change and confidence change is that the other major aspect of behaviour that improved with practice did so in a manner unrelated to clinical change. As noted above in the quoted text, participants’ discrimination improved from baseline to follow-up, reflected in the need for higher difficulty level to maintain accuracy around 70%. Crucially, this was not associated with symptom change. This speaks against a general mechanism where symptom improvement leads to increased practice effects in general. Only changes in confidence specifically are associated with improved symptoms. We have provided more detail on this in the Discussion (page 9, lines 324-326: “This association with clinical improvements was specific to metacognitive changes, and not changes in task performance, suggesting that changes in confidence do not merely reflect greater task familiarity at follow-up.”).

Relatedly, to what extent is there a role for general task engagement in these findings? The paper might be strengthened by some kind of control analysis, perhaps using (as a proxy for engagement) the data collected about those who missed catch questions in the questionnaires.

Response: Thank you for your comment. We included the details of data quality checks in the Supplement. Given the small number of participants that failed more than one attention checks (1% of the iCBT arm) and that all those participants passed the task exclusion criteria, we made the decision to retain these individuals for analyses. We have since examined if excluding these small number of individuals impacts our findings. Excluding those that failed more than one catch item did not affect the significance of results, which has now been added to the Supplementary Information (Data Quality Checks: Task and Clinical Scales, page 5, lines 181-185: “Additionally, excluding those that failed more than one catch item in the iCBT arm did not affect the significance of results, including the change in confidence (b=0.16, SE=0.02, p<0.001), change in anxious-depression (b=-0.32, SE=0.03, p<0.001), and the association between change in confidence and change in anxious-depression (r(638)=-0.10, p=0.011)”).

I was also unclear what the findings about task difficulty might mean. Are confidence changes purely secondary to improvements in task performance generally - so confidence might not actually be 'interesting' as a construct in itself? The authors could have commented more on this issue in the discussion.

Response: Thank you for this comment and sorry it was not clear in the original paper. As we discussed in a prior reply, accuracy – i.e. proportion of correct selections (the target of confidence judgements) are different from the difficulty of the dot discrimination task that each person receives on a given trial. We had provided more details on task difficulty in the Supplement. Accuracy was tightly controlled in this task using a ‘two-down one-up’ staircase procedure, in which equally sized changes in dot difference occurred after each incorrect response and after two consecutive correct responses. The task is more difficult when the dot difference between stimuli is lower, and less difficult when the dot difference between stimuli is greater. Therefore, task difficulty refers to the average dot difference between stimuli across trials. Crucially, task accuracy did not change from baseline to follow-up, only task difficulty. Moreover, changes in task difficulty were not associated with changes in anxious-depression, while changes in confidence were, indicating confidence is the clinically relevance construct for change in symptoms.

We appreciate that this may not have been clear from the description in the main manuscript, and have added more detail on task difficulty to the Methods (Metacognition Task, page 14, lines 540-542: “Task difficulty was measured as the mean dot difference across trials, where more difficult trials had a lower dot difference between stimuli.”) and Results (Treatment Findings: iCBT, pages 4-5, lines 184-186: “Although overall accuracy remained stable due to the staircasing procedure, participants’ ability to detect differences between the visual stimuli improved.”). We have also elaborated more on how improvements in symptoms are associated with change in confidence, not task performance in the Discussion (page 9, lines 324-326: “This association with clinical improvements was specific to metacognitive changes, and not changes in task performance, suggesting that changes in confidence do not merely reflect greater task familiarity at follow-up”).

To make code more reproducible, the authors could have produced an R notebook that could be opened in the browser without someone downloading the data, so they could get a sense of the analyses without fully reproducing them.

Response: Thank you for your comment. We appreciate that an R notebook would be even better than how we currently share the data and code. While we will consider using Notebooks in future, we checked and converting our existing R script library into R Notebooks would require a considerable amount of reconfiguration that we cannot devote the time to right now. We hope that nonetheless the commitment to open science is clear in the extensive code base, commenting and data access we are making available to readers.

Rather than reporting full study details in another publication I would have found it useful if all relevant information was included in a supplement (though it seems much of it is). This avoids situations where the other publication is inaccessible (due to different access regimes) and minimises barriers for people to fully understand the reported data.

Response: We agree this is good practice – the Precision in Psychiatry study is very large, with many irrelevant components with respect to the present study (Lee et al., BMC Psychiatry, 2023). For this reason, we tried to provide all that was necessary and only refer to the Precision in Psychiatry study methods for fine-grained detail. Upon review, the only thing we think we omitted that is relevant is information on ethical approval in the manuscript, which we have now added (Methods, Participants, page 11, lines 412-417: “Further details of the PIP study procedures that are not specific to this study can be found in a prior publication (21). Ethical approval for the PIP study was obtained from the Research Ethics Committee of School of Psychology, Trinity College Dublin and the Northwest-Greater Manchester West Research Ethics Committee of the National Health Service, Health Research Authority and Health and Care Research Wales”). If any further information is lacking, we are happy to include it here also.